# Improving oligo-conjugated antibody signal in multimodal single-cell analysis

Terkild B Buus[1,2,3]*, Alberto Herrera[1], Ellie Ivanova[1], Eleni Mimitou[4], Anthony Cheng[5,6], Ramin S Herati[7], Thales Papagiannakopoulos[1], Peter Smibert[4], Niels Odum[2,3], Sergei B Koralov[1]*

[1]Department of Pathology, New York University School of Medicine, New York, United States; [2]LEO Foundation Skin Immunology Research Center, University of Copenhagen, Copenhagen, Denmark; [3]Department of Immunology and Microbiology, University of Copenhagen, Copenhagen, Denmark; [4]Technology Innovation Lab, New York Genome Center, New York, United States; [5]Department of Genetics and Genome Sciences, University of Connecticut School of Medicine, Farmington, United States; [6]Department of Biostatistics and Epidemiology, School of Public Health and Health Sciences, University of Massachusetts, Amherst, United States; [7]NYU Langone Vaccine Center, Department of Medicine, New York University School of Medicine, New York, United States

**Abstract** Simultaneous measurement of surface proteins and gene expression within single cells using oligo-conjugated antibodies offers high-resolution snapshots of complex cell populations. Signal from oligo-conjugated antibodies is quantified by high-throughput sequencing and is highly scalable and sensitive. We investigated the response of oligo-conjugated antibodies towards four variables: concentration, staining volume, cell number at staining, and tissue. We find that staining with recommended antibody concentrations causes unnecessarily high background and amount of antibody used can be drastically reduced without loss of biological information. Reducing staining volume only affects antibodies targeting abundant epitopes used at low concentrations and is counteracted by reducing cell numbers. Adjusting concentrations increases signal, lowers background, and reduces costs. Background signal can account for a major fraction of total sequencing and is primarily derived from antibodies used at high concentrations. This study provides new insight into titration response and background of oligo-conjugated antibodies and offers concrete guidelines to improve such panels.

*For correspondence:
terkild.buus@sund.ku.dk (TBB);
sergei.koralov@nyulangone.org (SBK)

## Introduction

Analysis of surface proteins in multimodal single-cell genomics such as cellular indexing of transcriptomes and epitopes by sequencing (CITE-seq) is a powerful addition to conventional single-cell RNA sequencing (scRNA-seq) (*Stoeckius et al., 2017*; *Peterson et al., 2017*; *Mair et al., 2020*). Unlike flow- and mass cytometry, CITE-seq is not limited by spectral overlap or availability of distinguishable isotopes (*Gullaksen et al., 2019*; *Hulspas et al., 2009*). This is due to the practically unlimited number of distinct oligo barcodes and discrete sequence counting, allowing high numbers of antibodies to be included in individual experiments.

While signal acquisition in CITE-seq is different, the reagents and staining procedure are highly analogous to staining for flow cytometry. Traditional titration for flow or mass cytometry aims to identify the conjugated antibody concentration, allowing the best discrimination between the signal from positive and negative cells (*Gullaksen et al., 2019*; *Hulspas, 2010*). Multiple factors may affect antibody binding and subsequent signal including antibody concentration, total amount of antibody,

as well as the level of target expression (epitope amount). Epitope amount is governed by the number of cells and the per-cell expression of the target epitope. These factors are in turn influenced by the cellular composition of the sample as well as their activation and differentiation state. Nonspecific binding is expected to increase as the total amount of antibody molecules greatly exceeds the epitopes present in a sample. As such, nonspecific binding is dependent on the total number of antibody molecules, rather than the antibody concentration (*Hulspas et al., 2009*). This makes staining volume, cell composition, and cell number important parameters for optimal staining (*Hulspas, 2010*). Consequently, flow and mass cytometric optimization aims to use antibody concentrations that reach the highest signal-to-noise ratio (often reached at the 'saturation plateau') in a minimal volume (and thus minimal number of antibody molecules) (*Gullaksen et al., 2019*; *van Vreden, 2019*).

Oligo-conjugated antibody signal has been shown to be highly analogous to fluorochrome-conjugated antibodies of the same clone in flow cytometry in regards to the concentration needed to reach the 'saturation plateau' (*Stoeckius et al., 2018*). However, unlike flow cytometry, where antibody (fluorescence) signal intensity has no influence on analysis cost, oligo-conjugated antibody signal is analyzed by counting sequencing reads, making costs strictly dependent on signal intensity (by requiring increased sequencing depth). This is particularly important for methods sequencing vast numbers of cells stained with a high number of antibodies such as single-cell combinatorial indexed cytometry by sequencing (SCITO-seq), where shallow sequencing is paramount for the economic feasibility of such methods (*Hwang, 2020*). Thus, while an optimal antibody concentration in flow cytometry aims to get the highest signal-to-noise ratio, oligo-conjugated antibody staining conditions should be titrated to get sufficient signal-to-noise at the lowest possible signal intensity. In practice, this means that concentrations of most antibodies in an optimized CITE-seq panel are not intended to reach their 'saturation plateau', but should be within their linear concentration range (where doubling the antibody concentration leads to twice the signal). Such concentrations are much more sensitive to the number of available epitopes (i.e., cell number and cell composition) than an optimized flow cytometry panel. Unlike flow and mass cytometry, where the major source of background is autofluorescence, spillover between neighboring channels, and nonspecific binding of the antibodies (*Hulspas et al., 2009*; *Au-Yeung et al., 2019*), a major source of background signal for oligo-conjugated antibodies appears to be free-floating antibodies in the cell suspension (*Mulè et al., 2020*). In droplet-based single-cell sequencing methods, these free-floating antibodies will be distributed between cell-containing and empty droplets. As signal from empty droplets can only be distinguished from signal from cell-containing droplets after sequencing and due to the much higher number of empty than cell-containing droplets, background signal can make up a considerable fraction of the sequenced reads, and thus sequencing costs.

In this study, we present a limited but practically applicable titration of four variables in a 5′-CITE-seq panel of 52 antibodies: (1) antibody concentration (fourfold dilution response), (2) staining volume (50 µL vs. 25 µL), (3) cell count ($1 \times 10^6$ vs. $0.2 \times 10^6$), and (4) tissue of origin: peripheral blood mononuclear cells (PBMCs) from healthy donor vs. immune cell compartment from a lung tumor sample. We find that oligo-conjugated antibodies show high background and limited response to titration when used above 2.5 µg/mL and that most antibodies appear to reach their saturation plateau at concentrations between 0.62 and 2.5 µg/mL. Many antibodies can be further diluted, despite being at their linear concentration range, without affecting the identification of epitope-positive cells. Reducing staining volume has a minor effect on signal and only impacts signal from antibodies used at low concentrations targeting highly expressed epitopes; this effect is counteracted by reducing the number of cells present during staining. We compare samples stained with pre-titration and adjusted concentrations of the same antibody panel and find that adjusting concentrations increases signal, lowers background, and reduces both sequencing and antibody costs. Finally, we find that background signal in empty droplets can constitute a major fraction of the total sequencing reads and is skewed towards antibodies used at high concentrations targeting epitopes present in low amounts.

## Results

### Fourfold antibody dilution in PBMC and lung tumor immune cells

A panel of 52 oligo-conjugated antibodies was allocated into several groups of starting concentrations based on previous experience with each antibody, epitope abundance or following vendor recommendations (concentration range between 0.05 and 10 µg/mL; *Supplementary file 1*). We stained two samples of either $10^6$ PBMCs or $5 \times 10^5$ lung tumor leukocytes in 50 µL of antibody mixture with various starting concentrations, hereafter referred to as 'dilution factor (DF) 1'. To determine how the signal from each antibody changed by dilution across the two tissues, we stained the same number of cells in the same volume with a four times diluted antibody mixture (DF4).

Single-cell gene expression was assessed by shallow sequencing (~4000 reads per cell) to assign cells into major cell lineages (*Figure 1A*) and cell types (*Figure 1B*) based on their transcriptional profile (see *Figure 1—figure supplement 1* for gene detection and unique molecular identifier (UMI) distributions and details on cell-type annotation). Leukocytes from lung tumor samples exhibited distinct transcriptional profiles within each cell type, but showed overall good co-clustering with similar cell types (*Figure 1C*). To allow direct comparison of UMI counts from the different conditions, we reduced the number of cells included in analysis from each condition to contain the same number of cells from each cell type. By only using the gene expression modality for cell-type assignment, we can directly compare antibody-derived tag (ADT) UMI counts at different staining conditions within transcriptional subclusters without risk of having differences in ADT signal interfere with cell-type assignment.

Comparing the total ADT UMI counts from each condition, we saw fewer UMIs from samples stained with DF4 as compared with DF1, both at 77% sequencing saturation (*Figure 1D*). However, the reduction in UMI counts from DF1 to DF4 by 38% (761,350 to 474,404) and 51% (1,121,940 to 548,393) in PBMC and lung, respectively, was markedly less than the fourfold difference (75% reduction) in antibody concentrations used in staining. It is worth noting that 4/52 antibodies used at the highest concentration (10 µg/mL) accounted for more than 20% of the total UMI counts irrespective of tissues and dilution factors and without showing any clearly positive populations (*Figure 1D, E*; gating thresholds shown in *Figure 1—figure supplement 2*). Indeed, we found that the majority of antibodies used in concentrations at or above 2.5 µg/mL showed minimal response to fourfold titration, both in terms of total UMI counts (*Figure 1F*) as well as UMI counts at the 90th quantile of the cell type with the highest overall expression level (*Figure 1G*; expressing cell types identified in *Figure 1E*), reflecting the response within the positive population where such could be identified. In contrast, antibodies used in concentrations at or below 0.62 µg/mL all showed close to linear response to fourfold dilution (shown as a reduction around two 'logs' on a log2 scale; *Figure 1F, G*). This indicates that the signal for many antibodies reach their saturation plateau in the range between 0.62 and 2.5 µg/mL, and that higher concentrations are likely to only increase the background signal.

In the present antibody panel, the response to fourfold dilution can be divided into five categories (*Figure 2*, *Figure 2—figure supplements 1–5*) that warrant different considerations in the choice of whether to reduce concentration or not. For category A (*Figure 2A*), reducing concentration is always the right choice. For the other categories (*Figure 2B–E*), the choice of whether to reduce concentration or not depends on the balance between the need for signal and the economic cost of signal (see *Table 1*).

### Reducing staining volume primarily affects highly expressed markers

To investigate the effect on ADT signal caused by further reducing the staining volume, we included PBMC samples stained with the same concentration of antibodies in 50 µL or 25 µL (effectively using half the amount of antibodies at twice the cell density). In both samples, we used the DF4 panel on $10^6$ cells to assess the worst-case scenario of the reduction as the amount of epitopes in this setting is likely to be competing for antibodies that are not in excess. Despite having many antibodies responding linearly to concentration reduction (*Figure 1*), we found much less response to reduced staining volume, both in regard to total number of UMIs (9% reduced; 469,541 to 428,680) and on a marker by marker basis (*Figure 3A–C*). As expected, antibodies used in low concentrations (0.0125– 0.025 µg/mL) targeting highly abundant epitopes were most severely affected by the reduced

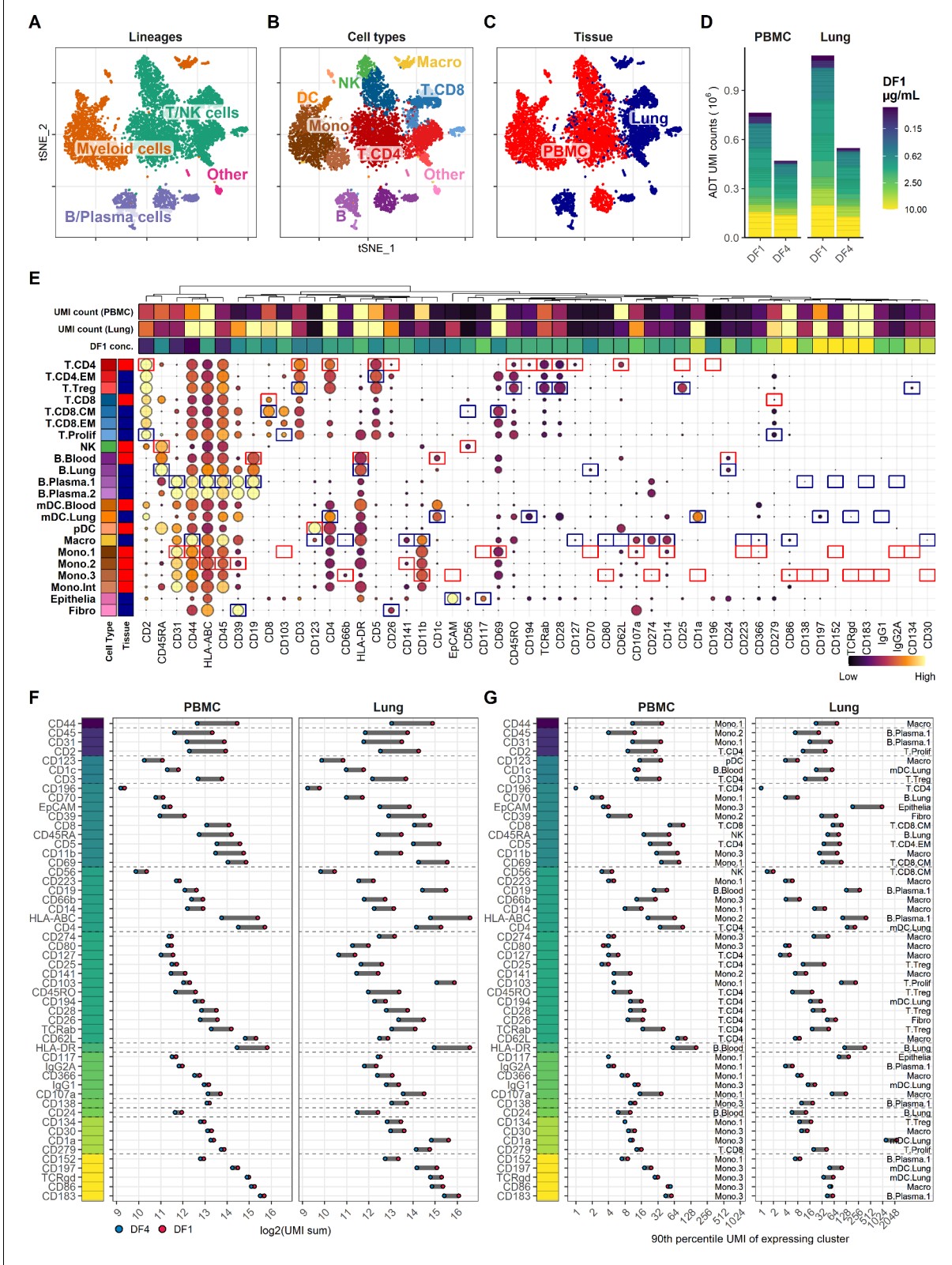

**Figure 1.** Fourfold antibody dilution response in peripheral blood mononuclear cell (PBMC) and lung tumor immune cells. (A–C) Single cells from all samples and conditions were clustered and visualized according to their gene expression and colored by (A) overall cell lineage, (B) cell type, and (C) tissue of origin. (D) Summarized unique molecular identifier (UMI) counts within cell-containing droplets segmented by the individual antibodies stained at the starting concentrations (dilution factor 1 [DF1]) or at a fourfold dilution (DF4) in PBMC and lung samples (concentrations of each antibody can be

*Figure 1 continued on next page*

*Figure 1 continued*

found in *Supplementary file 1*). Antibody segments are colored by their concentration at DF1. (E) Heatmap of normalized antibody-derived tag (ADT) signal within each transcription-based cluster identified in (B). Visualized by frequency of positive cells (circle size) and colored by the median ADT signal within the positive fraction (i.e., signal from a marker that is highly expressed by all cells in a cluster will have the biggest circle and be colored yellow). Red and blue colored boxes denote the clusters chosen for evaluating titration response within blood and lung samples, respectively. (F, G) Change in ADT signal for each antibody by fourfold dilution. Individual antibodies are colored by their concentration at DF1 and quantified by (F) sum of UMIs within cell-containing droplets assigned to each antibody and (G) 90th percentile UMI count within expressing cell cluster identified in (E) and annotated by numbers to the right.

The online version of this article includes the following figure supplement(s) for figure 1:

**Figure supplement 1.** Quality control metrics and cell-type annotation.

**Figure supplement 2.** Gating positive cells based on antibody-derived tag (ADT) signal at dilution factor 1.

staining volume (such as CD31, CD44, and CD45; *Figure 3D, E*, *Figure 3—figure supplements 1* and *2*), whereas antibodies targeting less abundant epitopes were largely unaffected (such as CD8 and CD19; *Figure 3F*).

## Reducing cell number during staining increases signal for antibodies at low concentration

To determine if the limited effect of reduced staining volume on ADT signal could be counteracted by simultaneously reducing the number of cells at the time of staining (effectively reducing the total amount of epitopes), we analyzed two PBMC samples with either $1 \times 10^6$ or $0.2 \times 10^6$ cells stained with the same concentration of antibodies (DF4) in 25 µL. Similar to reducing staining volume, the majority of the included antibodies were largely unchanged by lowering the cell density at staining, as reflected by only 8% increase in detected UMIs (from 428,680 to 462,916), and also reflected by the analogous distribution of individual markers (*Figure 4A–C*). Encouragingly, reducing the cell number at staining increased the signal from the antibodies used at low concentration and targeting highly expressed epitopes (*Figure 4D, E*, *Figure 4—figure supplements 1* and *2*), thus largely mitigating the loss of signal observed when the staining volume was reduced from 50 µL to 25 µL (*Figure 3B–D*). Interestingly, despite reducing the cell density at staining fivefold (from 40 to $8 \times 10^6$ cells/mL), the resulting signal did not appreciably surpass that of the sample stained in 50 µL with an intermediate cell density of $20 \times 10^6$ cells/mL (*Figure 4—figure supplement 3*).

## Adjusting antibody concentration improves signal, lowers background, and reduces cost and sequencing requirements

To evaluate the benefits of adjusting antibody concentrations, we stained 200,000 PBMCs in a staining volume of 25 µL using the same antibody panel, with individual antibody concentrations adjusted based on their assigned categories (individual concentrations can be found in *Supplementary file 1,* and how each category was adjusted is described in *Table 1*). On average, the adjusted panel used 1.9-fold less antibody than the DF1 staining and 8.4-fold less than the vendor-recommended starting concentration (*Supplementary file 2*). Together with the reduced staining volume, this decreased antibody costs per sample to 50 USD, which is a 3.9- and 33.6-fold reduction from DF1 (195 USD) and vendor recommendations (1690 USD), respectively (based on list price of 325 USD per 10 µg; *Supplementary file 2*).

To allow direct comparison with the DF1 sample, we integrated and down-sampled the DF1 and adjusted samples to include similar numbers cells within each cell type (*Figure 5A*). We then down-sampled the sequenced ADT reads to yield similar UMI totals of 522,469 and 521,331 across the comparable cell populations for the DF1 and adjusted sample, respectively (*Figure 5B*). As expected, antibodies used at reduced concentrations yielded relatively fewer UMIs (categories A and B and some from E), whereas increased concentrations yielded more (category E and some from C). Importantly, we found that antibodies with unchanged concentration yielded more UMIs at similar sequencing depth (*Figure 5B, C*). This was primarily due to a reduction of category A antibodies that accounted for 25% of the sequenced UMI sequences in the DF1 sample and only 10% in the adjusted sample.

Due to the cost of signal in these sequencing-based approaches, an optimal panel would ideally use similar number of UMIs per positive cell for each antibody (*Figure 5D*) and exhibit approximately

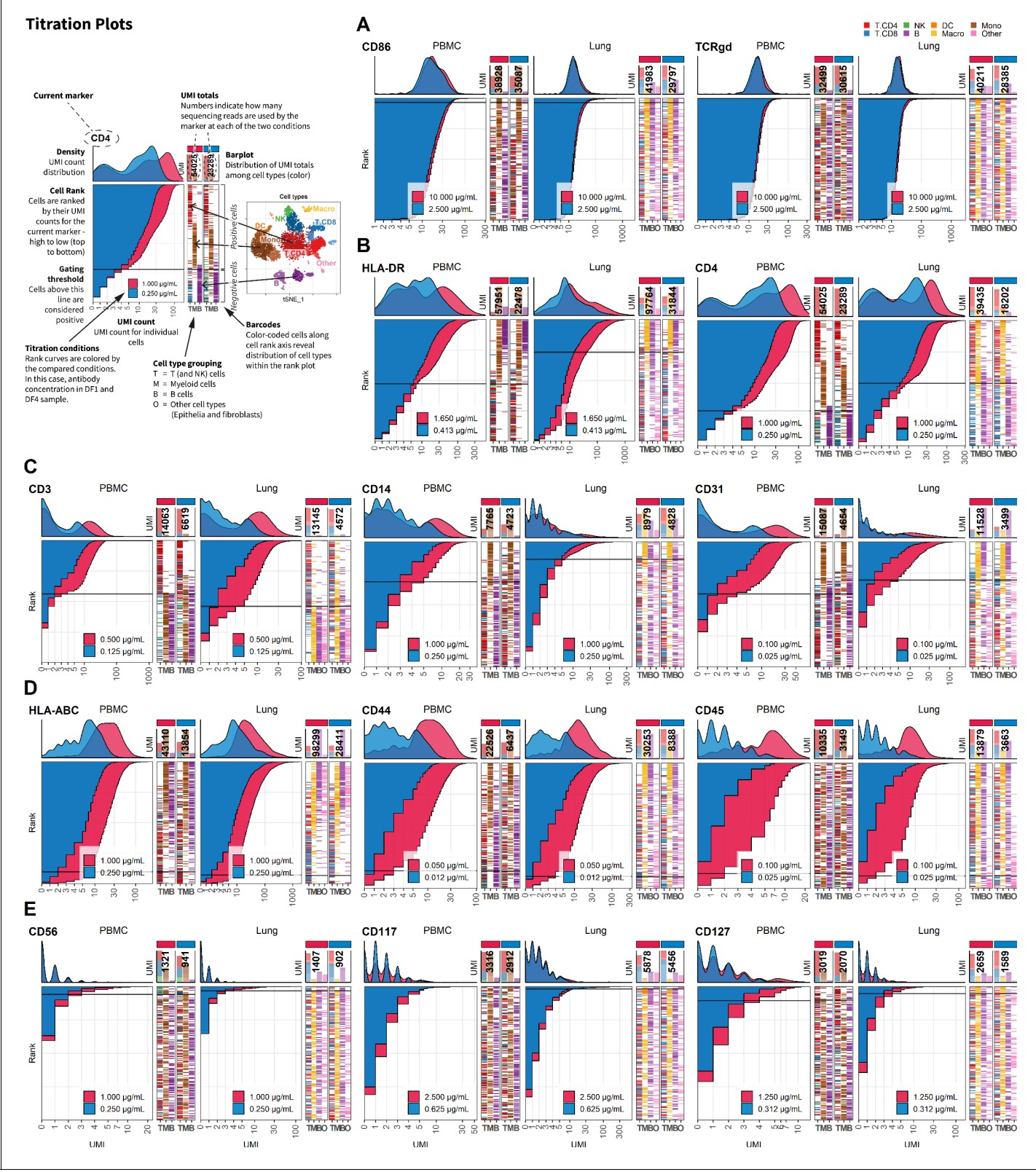

**Figure 2.** Fourfold antibody dilution response is dependent on epitope abundance. Titration plots (unique molecular identifier [UMI] count vs. cell rank) showing response to reduction in antibody concentration from dilution factor 1 (DF1) to DF4 within peripheral blood mononuclear cells (left) and lung (right). Histogram depicts distribution of UMIs at each condition colored by dilution factor (and annotated with concentration). Numbers within bar plot denote total UMI count within cell-containing droplets at each antibody concentration. Barcodes to the right depict cell type by color at the

*Figure 2 continued on next page*

*Figure 2 continued*

corresponding rank to visualize specificity of the antibody. Horizontal line depicts gating threshold for cells considered positive for the marker. Antibody response to fourfold dilution can be divided into five categories exemplified in (A-E). (A) Antibodies where the positive signal is obscured within the background signal (category A). (B) Antibodies that respond by a reduction in signal but without hampering the ability to distinguish positive from negative cells (category B). These antibodies also show strict cell-type specificity (i.e., HLA-DR is restricted to non-T cells, whereas CD4 is highly expressed in T cells and intermediately expressed in myeloid cells as shown in the barcode plot). (C) Antibodies that respond by a reduction in both signal and change the ability to distinguish positive from negative cells (category C). (D) Antibodies targeting ubiquitously expressed markers (category D). (E) Antibodies that do not show a convincing positive population due to either lack of epitopes (no positive cells in either tissue) or lack of antibody binding (non-functional antibody) (category E). Titration plots for all markers can be found in *Figure 2—figure supplements 1–5*.

The online version of this article includes the following figure supplement(s) for figure 2:

**Figure supplement 1.** Response of individual antibodies to fourfold reduction in concentration in peripheral blood mononuclear cells (PBMCs) and lung tumor immune cells – category A.

**Figure supplement 2.** Response of individual antibodies to fourfold reduction in concentration in peripheral blood mononuclear cells (PBMCs) and lung tumor immune cells – category B.

**Figure supplement 3.** Response of individual antibodies to fourfold reduction in concentration in peripheral blood mononuclear cells (PBMCs) and lung tumor immune cells – category C.

**Figure supplement 4.** Response of individual antibodies to fourfold reduction in concentration in peripheral blood mononuclear cells (PBMCs) and lung tumor immune cells – category D.

**Figure supplement 5.** Response of individual antibodies to fourfold reduction in concentration in peripheral blood mononuclear cells (PBMCs) and lung tumor immune cells – category E.

**Table 1.** Five categories of response to fourfold dilution.

| Categories | Responses to fourfold dilution | Markers | Considerations |
|---|---|---|---|
| A (*Figure 2A*) | Antibodies exhibiting no response to dilution, indicating that the positive signal is fully saturated, absent, or obscured within high background signal. | *CD1a, CD30, CD86, CD134, CD138, CD152, CD183, CD197, CD279, CD336, IgG1, IgG2A,* and *TCRgd* | Reducing antibody concentration is always the right choice. These antibodies sequester a large amount of unique molecular identifiers without yielding critical insight. Reducing concentration may reveal a true positive population obscured by the background signal. |
| B (*Figure 2B*) | Antibodies that respond by a reduction in signal but without hampering the ability to distinguish positive and negative fractions. | *CD4, CD5, CD8, CD11b, CD19, CD62L, CD69, CD103, CD107a, CD194, CD274, EpCAM, HLA-DR,* and *TCRab* | Reducing antibody concentrations will be economically beneficial with minimal loss of biological information. For instance, In the lung at dilution factor 1, HLA-DR uses 9% of the total unique molecular identifier counts within cell-containing droplets and can be reduced at least fourfold without any apparent change in ability to discriminate between positive and negative cells. |
| C (*Figure 2C*) | Antibodies that respond by a reduction in signal that subsequently changes the ability to distinguish positive from negative cells or bring the cutoff value for positive cells down to only a few unique molecular identifiers. | *CD1c, CD2, CD3, CD14, CD25, CD26, CD28, CD31, CD39, CD45RA, CD45RO,* and *CD141* | Reducing antibody concentration will reduce biological information as cells expressing the targeted epitopes may not exhibit sufficient signal. If only cells expressing high levels of the marker need to exhibit signal, these can be slightly reduced. |
| D (*Figure 2D*) | Antibodies that respond linearly to titration but take up high numbers of unique molecular identifiers due to targeting (almost) ubiquitously expressed markers. | *CD44, CD45,* and *HLA-ABC* | These can be reduced if all cells exhibit high unique molecular identifier counts. Unless these markers have a clear purpose, most experiments will benefit from dropping them from the panel as they tend to sequester a large proportion of total sequencing reads with little biological information. |
| E (*Figure 2E*) | Antibodies where response is hard to assess due to not showing expected positive population either due to lack of epitopes (no positive cells in either tissue) or lack of antibody binding (non-functional antibody). | *CD24, CD56, CD66b, CD70, CD80, CD117, CD123, CD127, CD196,* and *CD223* | Should be evaluated individually. Is there prior information indicating that this marker is expressed by cells in these types of samples? Do any cells in the sample express high levels of the gene encoding the targeted protein? If so, increasing the concentration of the antibody or trying a different clone may yield better signal. |

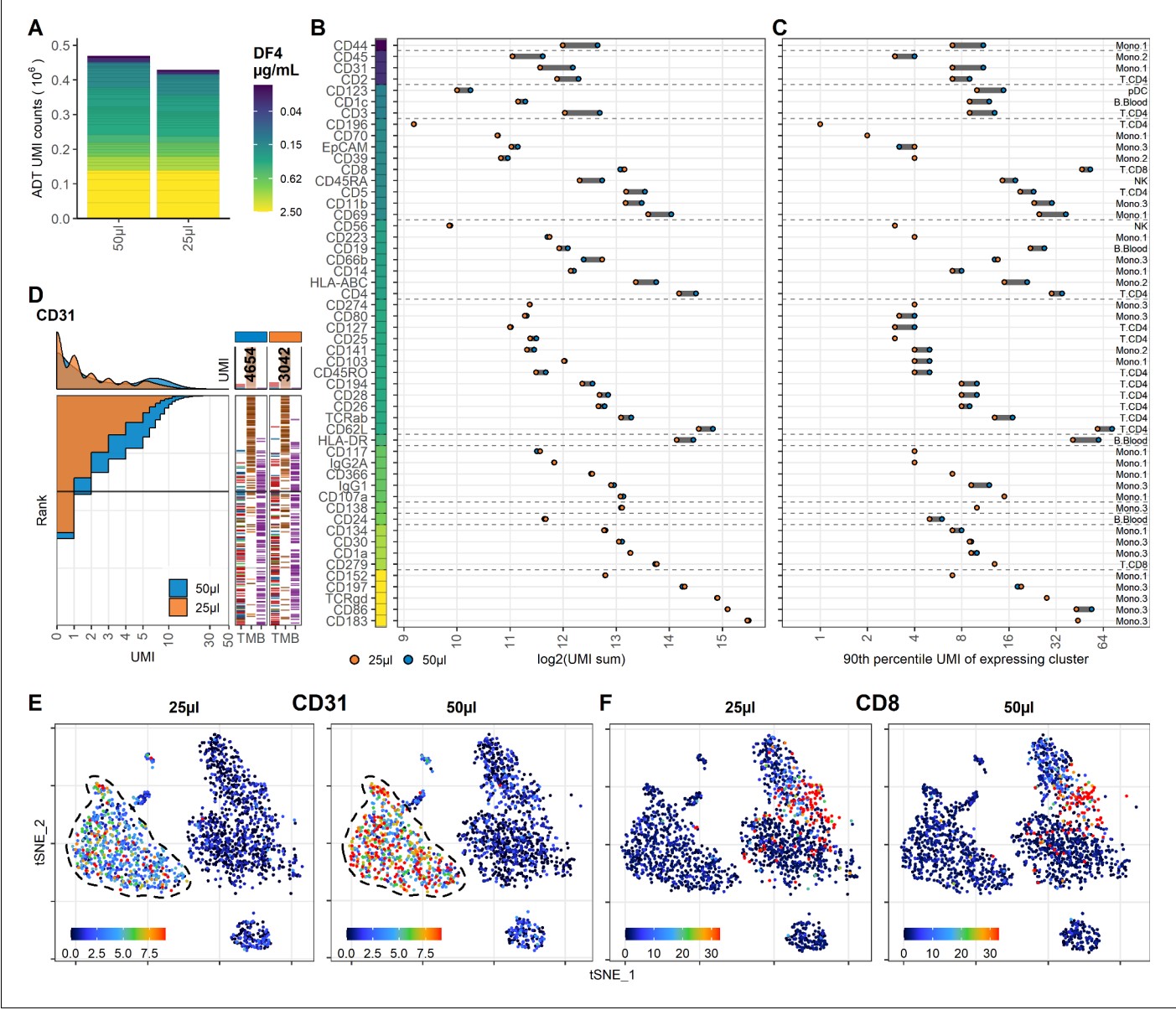

**Figure 3.** Reducing staining volume primarily affects highly expressed markers. Comparison of peripheral blood mononuclear cell samples stained in 50 µL (same sample as dilution factor [DF] 4 in *Figure 1*) or 25 µL volume at DF4. (A) Summarized unique molecular identifier (UMI) counts within cell-containing droplets segmented by the individual antibodies colored by their concentration. (B, C) Change in antibody-derived tag signal for each antibody by reducing staining volume from 50 to 25 µL. Individual antibodies are colored by their concentration. Quantified by (B) sum of UMIs within cell-containing droplets assigned to each antibody and (C) 90th percentile UMI count within the cell type with most abundant expression (the assayed cell type is annotated to the right). (D) Titration plot (marker UMI count vs. normalized cell rank) for CD31 signal response when reducing staining volume from 50 µL to 25 µL. Histogram depicts distribution of UMIs at each condition. Barcodes to the right depict cell-type occurrence at the corresponding rank to visualize cell specificity of the antibody. Numbers on top of the small bar plot denote total UMI count assigned to CD31 within cell-containing droplets from each condition. (E, F) Non-normalized UMI counts visualized on t-distributed stochastic neighbor embedding (tSNE) plots of an affected (CD31; E) or an unaffected (CD8; F) marker by the reduction in cell density. Dashed line indicates the region where expression levels vary between volumes. Titration plots for all markers can be found in *Figure 3—figure supplements 1* and *2*.

The online version of this article includes the following figure supplement(s) for figure 3:

**Figure supplement 1.** Response of individual antibodies to reduction in staining volume.

**Figure supplement 2.** Response of individual antibodies to reduction in staining volume.

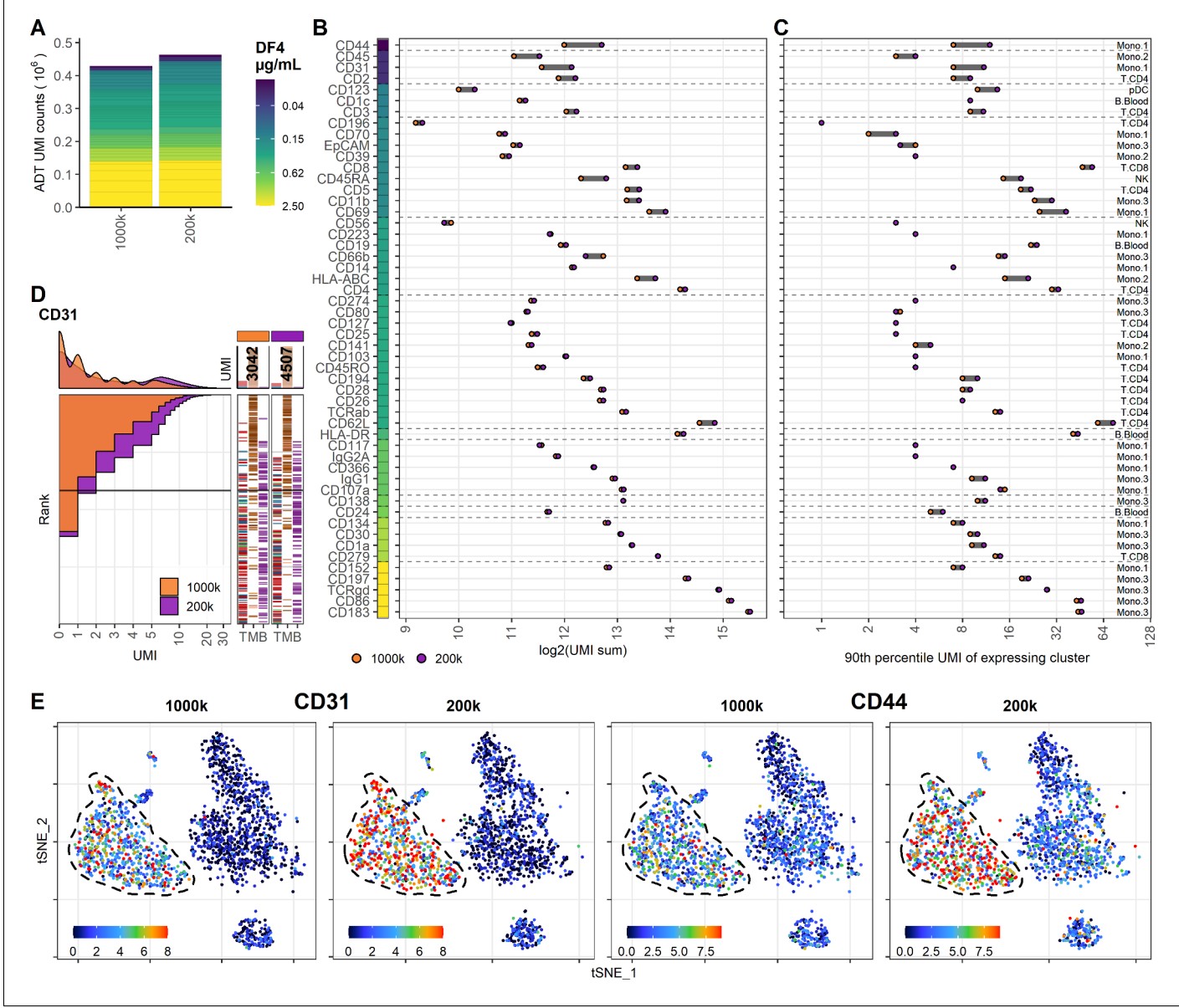

**Figure 4.** Reducing cell number during staining increases signal for antibodies at low concentration. Comparison of peripheral blood mononuclear cell samples stained in 25 μL antibody staining solution at dilution factor 4 at two cell densities: $1 \times 10^6$ (1000k; same sample as 25 μL in *Figure 3*) or $0.2 \times 10^6$ (200k) cells. (A) Summarized unique molecular identifier (UMI) counts within cell-containing droplets segmented by the individual antibodies colored by their concentration. (B, C) Change in antibody-derived tag signal for each antibody by reducing cell numbers at staining from $1 \times 10^6$ to $0.2 \times 10^6$ cells. Individual antibodies are colored by their concentration. Quantified by (B) sum of UMIs within cell-containing droplets assigned to each antibody and (C) 90th percentile UMI count within cell type with most abundant expression (the assayed cell type is annotated to the right). (D) Titration plot (marker UMI count vs. normalized cell rank) for CD31 signal response when reducing cell numbers at staining from $1 \times 10^6$ to $0.2 \times 10^6$ cells. Histogram depicts distribution of UMIs at each condition. Barcodes to the right depict cell-type occurrence at the corresponding rank to visualize cell specificity of the antibody. Numbers on top of the small bar plot denote total UMI count assigned to CD31 within cell-containing droplets from each condition. (E) Non-normalized UMI counts visualized on t-distributed stochastic neighbor embedding (tSNE) plot of CD31 and CD44 that are affected by the reduction in staining volume, mitigated by a concomitant reduction in cell density. Dashed line indicates the region where expression levels vary between cell densities. Titration plots for all markers can be found in *Figure 4—figure supplements 1* and *2*.

The online version of this article includes the following figure supplement(s) for figure 4:

**Figure supplement 1.** Response of individual antibodies to reduction in cell numbers at staining.

**Figure supplement 2.** Response of individual antibodies to reduction in cell numbers at staining.

**Figure supplement 3.** Fivefold reduction in cell density mitigates but does not supersede twofold reduction in staining volume.

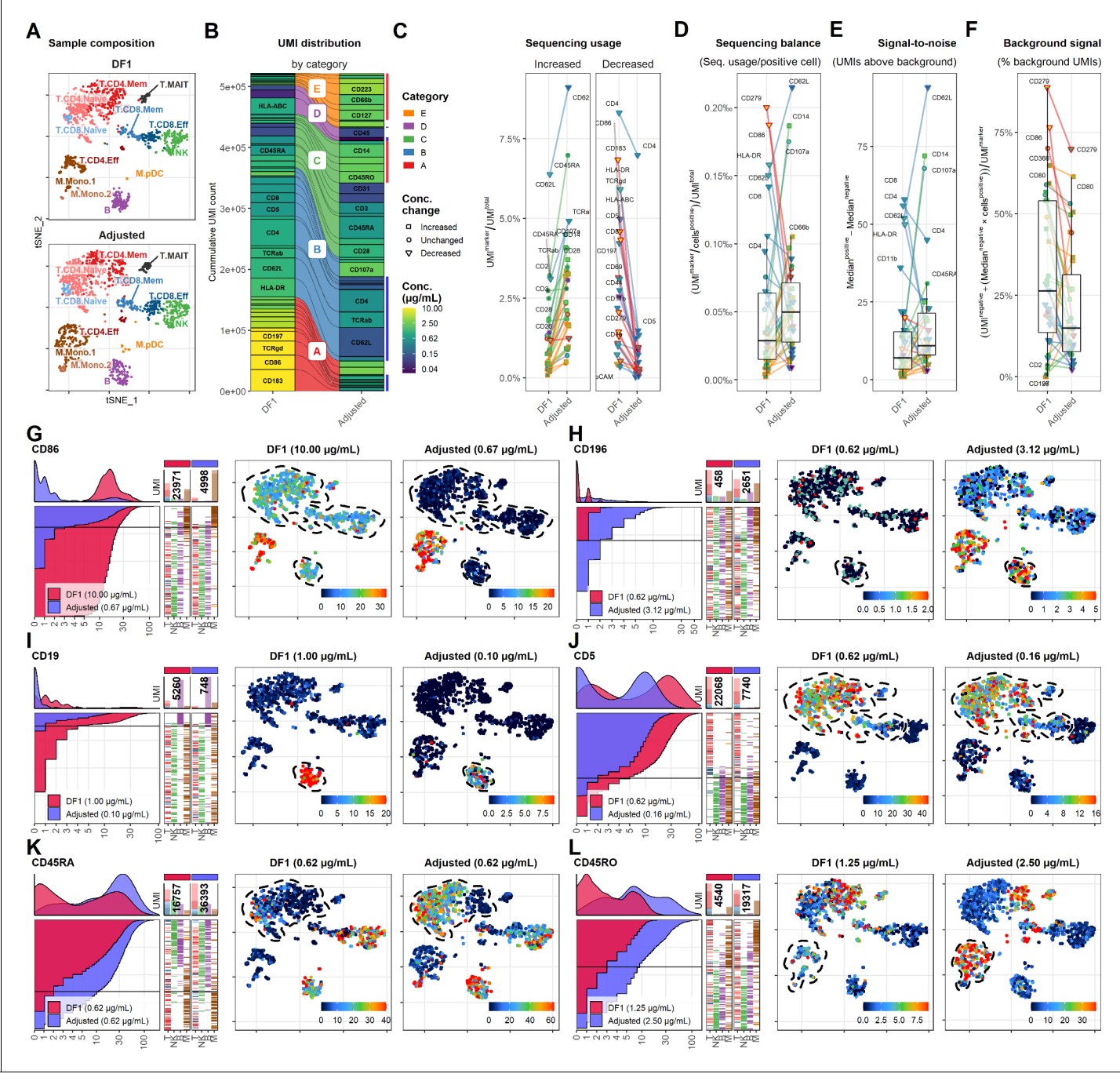

**Figure 5.** Adjusting antibody concentrations increases signal, lowers background, and reduces costs and sequencing requirements. (**A**) Single cells from the dilution factor (DF) 1 and adjusted sample were integrated and selected to yield similar number of cells within each annotated cell type, visualized by t-distributed stochastic neighbor embedding (tSNE). (**B**) Antibody-derived tag reads from DF1 and adjusted samples were subsampled to yield similar number of unique molecular identifiers (UMIs) within the selected cells. Size of each segment shows the distribution of UMIs among the antibodies in the panel divided into categories that determined how they were adjusted (**Table 1**). (**C–F**) Response of adjustment of individual antibodies assayed by (**C**) their overall sequencing usage (fraction of UMIs assigned to each marker), (**D**) balancing (percent of UMIs used per positive cell), (**E**) signal-to-noise (difference in median UMI count within positive and negative cells), and (**F**) background signal (percentage of UMIs used for background signal). Shapes of marker denote whether the antibody concentration was changed between the DF1 and adjusted sample. Color of 'shapes' denotes antibody concentration. Color of connecting lines denotes antibody category. Center line in box plot denotes the median. (**G–L**) Titration plot (left) and tSNE plots showing raw UMI counts (right) for antibodies in different categories. Dashed lines indicate regions of interest highlighting the differences (or lack thereof) between the DF1 and adjusted samples. Titration plots for all markers by category can be found in *Figure 5—figure supplements 1–5*.

*Figure 5 continued on next page*

*Figure 5 continued*

The online version of this article includes the following figure supplement(s) for figure 5:

**Figure supplement 1.** Dilution factor (DF) 1 vs.adjusted antibody concentration comparisons – category A.
**Figure supplement 2.** Dilution factor (DF) 1 vs. adjusted antibody concentration comparisons – category B.
**Figure supplement 3.** Dilution factor (DF) 1 vs. adjusted antibody concentration comparisons – category C.
**Figure supplement 4.** Dilution factor (DF) 1 vs. adjusted antibody concentration comparisons – category D.
**Figure supplement 5.** Dilution factor (DF) 1 vs. adjusted antibody concentration comparisons – category E.

the same positive signal (UMIs above background; *Figure 5E*). While some markers should be further reduced (such as CD4, CD45RA, CD62L, CD107a, and TCRab) and some adjustments were too extreme (such as CD14, CD19, and HLA-ABC), the adjusted sample exhibited close to a twofold increase in the median UMIs per positive cell and a 57% increase in the median positive signal (from 7 to 11 UMIs; *Figure 5E*). Importantly, all the markers with the lowest positive signal as well as number of UMIs per positive cell were all increased, reflecting a more balanced sequencing library.

Importantly, while exhibiting approximately the same relative background signal as assayed by proportion of reads within empty droplets (35–45%; data not shown), the adjusted sample generally showed much lower percentage of UMIs being assigned to background (*Figure 5F*). This was particularly remarkable for CD86, which went from 76.5% to 12.6% and thus yielded similar positive signal while using 4.8-fold fewer UMIs (from 23,971 to 4998; *Figure 5G*). In fact, the exception to this was primarily found within category E antibodies for which concentrations were increased due to having very low UMI counts in the DF1 sample (CD56, CD127, and CD196; see *Figure 5—figure supplements 1–5* for data on all markers). In these cases, the increased concentration yielded better definition of expected positive populations (*Figure 5H*). To balance the sequencing requirements of the panel, we reduced concentrations of most category B antibodies. Except CD19 (*Figure 5I*), all reduced category B antibodies showed no change in resolution of positive vs. negative populations despite a marked reduction in their UMI usage (*Figure 5C, D*) and concomitant reduction in their positive signal. For instance, when reducing anti-CD5 from 0.62 to 0.16 µg/mL, it showed largely identical distribution despite using 65% less UMIs (from 22,068 to 7740; *Figure 5J*). Category C and E antibodies showed consistently increased positive signal (*Figure 5D*) and consequently allowed better identification of populations known to express these markers, such as naïve T cells and monocytes for CD45RA and CD45RO, respectively (*Figure 5K, L*).

## Background signal from oligo-conjugated antibodies is dependent on antibody concentration and abundances of epitopes

Free-floating antibodies in the solution have been shown to be one of the major contributors to background signal for ADTs (*Mulè et al., 2020*). Similar to cell-free RNA, background ADT signal can be assayed from empty droplets. To determine the background signal of the different antibodies in our panel, we split the captured barcodes into cell-containing and empty droplets based on the inflection point of the barcode-rank plot for the gene expression UMI counts (*Figure 6—figure supplement 1*). Despite being a 'super-loaded' 10X Chromium run targeting 20,000 cells, the number of empty droplets vastly outnumbered the cell-containing droplets. Consequently, several antibodies exhibited more cumulated UMIs within empty droplets than within cell-containing droplets (*Figure 6A*). This was particularly prevalent within antibodies used at concentration of or above 2.5 µg/mL, thus drastically skewing the frequency of these antibodies within the empty droplets as compared with cell-containing droplets (*Figure 6A, B*). Conversely, antibodies targeting highly abundant epitopes were enriched within cell-containing droplets, irrespective of their staining concentration (such as CD44 and CD107a, HLA-ABC, HLA-DR; *Figure 6C*). This was consistent with publicly available datasets where ADTs from antibodies targeting abundant epitopes (such as CD3, CD4, CD8, and CD45RA) were enriched within the cell-containing droplets using two different capture approaches (3'- and 5' capture; *Figure 6—figure supplement 2*). We found that ADT signal in empty droplets (i.e., background) was highly correlated with the UMI cutoff for detection (*Figure 6D*, *Figure 6—figure supplements 3* and *4*). Markers with low background generally showed low UMI cutoff and exhibited high dynamic range, allowing identification of multiple levels of expression (as seen for CD4 and CD19; *Figure 6D, E*). In contrast, markers with high background

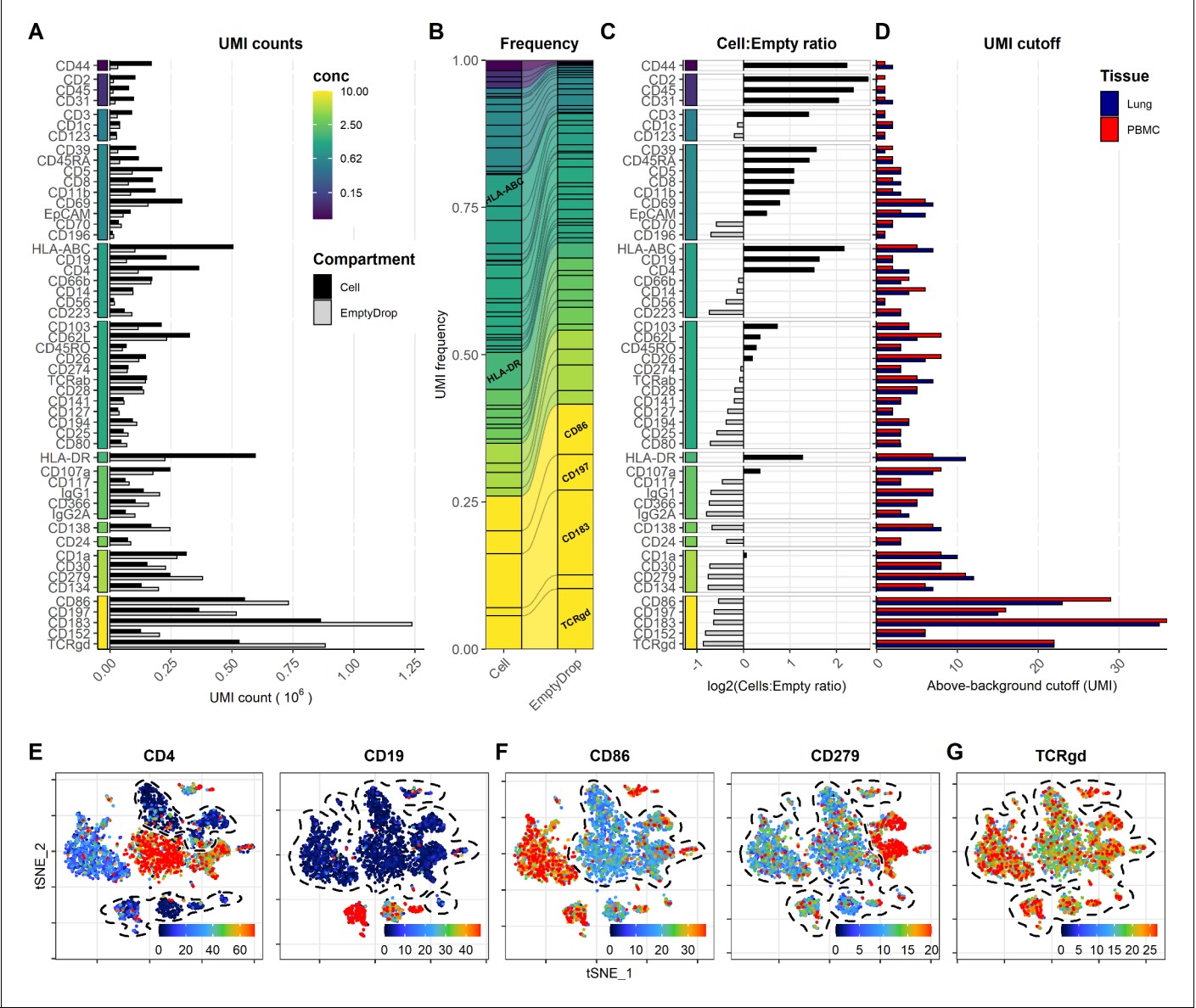

**Figure 6.** Background signal from oligo-conjugated antibodies is dependent on concentration and presence of epitopes. Signal from free-floating antibodies in the cell suspension is a major source of background in droplet-based scRNA-seq and can be assayed by their signal within non-cell-containing (empty) droplets. (A, B) Comparison of signal from each antibody within cell-containing and empty droplets (identified in *Figure 6—figure supplement 1*) by (A) their total unique molecular identifier (UMI) counts or (B) their relative frequency within each compartment. Color bar denotes antibody concentration at dilution factor 1 (DF1). (C) Ratio of UMI frequencies of each marker between cell-containing and empty droplets. Markers with black bars have greater frequency in cell-containing droplets, whereas gray bars have greater frequency in empty droplets. (D) UMI thresholds for detection above-background for each marker within peripheral blood mononuclear cells and lung tumor samples (based on gating in *Figure 1—figure supplement 2*). (E–G) Examples of t-distributed stochastic neighbor embedding (tSNE) plots showing non-normalized (raw) UMI counts from cells stained at DF1 for (E) markers with low background, (F) markers with high background that still exhibit cell-type-specific signal (CD86 and CD279), and (G) markers where positive signal is absent or obscured by the background. Regions of background signal are encircled by dashed lines. To make the color scale in the tSNE plots less sensitive to extreme values, we set the upper threshold to the 90% percentile. tSNE plots for all markers can be found in *Figure 6—figure supplements 3* and *4*.

The online version of this article includes the following figure supplement(s) for figure 6:

**Figure supplement 1.** Quantifying unique molecular identifiers (UMIs) within cells and empty droplets of antibody-derived tag (ADT) and hashtag-oligo (HTO).

**Figure supplement 2.** Quantifying unique molecular identifiers (UMIs) within cell-containing and empty droplets from public 10X datasets.

**Figure supplement 3.** Cellular distribution of ADT signal visualized by t-distributed stochastic neighbor embedding (tSNE) plots displaying raw (unnormalized) UMI counts from the cells stained at dilution factor 1 for each antibody.

*Figure 6 continued on next page*

Figure 6 continued

**Figure supplement 4.** Cellular distribution of ADT signal visualized by t-distributed stochastic neighbor embedding (tSNE) plots displaying raw (unnormalized) UMI counts from the cells stained at dilution factor 1.

showed high UMI cutoff regardless of whether they exhibited cell-type-specific signal (such as CD86 and CD279; *Figure 6F*) or whether their positive signal was absent or obscured by the high background (such as TCRγδ; *Figure 6G*).

## Discussion

In this study, we show that titration of oligo-conjugated antibodies for multimodal single-cell analysis can improve sensitivity, lower background signal, and reduce costs and sequencing requirements, and that such optimizations go beyond (and even against) the need to reach the 'saturation plateau'. We show that for a representative panel of 52 antibodies, most antibodies used in concentrations at or above 2.5 µg/mL show high background signal and we observed minimal loss in sensitivity upon a fourfold reduction in concentration of these antibodies. Antibodies used at concentrations between 0.625 and 2.5 µg/mL show limited (nonlinear) response, whereas most antibodies used at concentrations below 0.625 µg/mL show linear or close to linear response. It should be noted that these estimates may be inherently biased given that the starting concentrations were based on our prior experience with the individual antibody clones and our assumptions regarding abundance of targeted epitopes. This has favored using higher concentrations for antibodies known to have low performance and for antibodies with unknown performance. Nonetheless, for antibodies with unknown performance, our results highlight the benefits of conducting titration experiments or initially using the antibodies at concentrations in the 0.625–2.5 µg/mL range, rather than the 5–10 µg/mL range recommended by published antibody staining protocols and by commercial vendors. This is particularly important when adding new antibodies to existing panels, where antibodies added in a high concentration may account for a disproportionate usage of the total sequencing reads without providing any biological information (as seen for CD86, CD152, CD183, CD197, and TCRγδ in the DF1 panel). Our results also show that concentrations of antibodies targeting highly expressed epitopes can be further reduced without affecting resolution of positive and negative cells, even when these antibodies are already used within their linear concentration range (such as CD5, CD8, and CD19). By reducing the concentration of these antibodies, the allocation of reads to each antibody becomes more balanced between epitopes present at disparate abundance, allowing the overall sequencing depth to be reduced and maximizing the yield of a sequencing run.

By using varying starting concentrations based on prior experience and titrating the full panel together, our study does not necessarily identify optimal concentrations of individual antibodies. This could have been achieved by using saturating starting concentrations and additional serial dilutions, as has been previously done for a few markers (*Stoeckius et al., 2018*). However, due to the cost of signal in these cytometry-by-sequencing methods, using all antibodies at their highest signal-to-noise ratios would require much deeper sequencing as highly expressed markers would use the vast majority of the total sequencing reads. Instead, we aimed to get sufficient signal-to-noise, while keeping the sequencing allocated to each marker balanced. A further complication for titration experiments that start with saturating amounts of antibody is the observation that background signal can be largely attributed to free-floating antibodies in the solution. Thus, using high concentrations for all markers in one or more sample would increase the background in all samples if these were multiplexed into the same droplet segregation. This would likely obscure the positive signals and possibly titration response at lower concentrations (similar to what we see for category A antibodies). To avoid this, each condition would have to be run in its own droplet segregation, making traditional titration experiments prohibitively costly.

In this study, we used commercially available antibody clones that have been extensively used for other applications such as flow and mass cytometry, and we do see high concordance between ADT signals and the expected antigens within each cell type. Our approach did not allow us to formally test whether each antibody is specific to its intended antigen as we inferred specificity based on our understanding of the included cell types and looked for concordance with gene expression signature

of the cells. However, it should be noted, that when using antibody clones that are unfamiliar or have not undergone extensive testing, it is important to assure their specificity.

Reducing staining volume for $10^6$ PBMCs from 50 µL to 25 µL only showed a minor effect on signal, and this minimal impact was primarily observed for antibodies used at very low concentrations (0.0125–0.025 µg/mL) targeting highly expressed epitopes (such as CD31, CD44, and CD45). This effect was readily counteracted by concomitantly reducing the number of cells at staining to 0.2 × $10^6$ PBMCs in 25 µL. In flow cytometry, while the binding of antibody is strictly dependent on its concentration, background signal is dependent on the ratio between the total amounts of antibody and epitopes (*Hulspas, 2010*). Consequently, background can be reduced by increasing the number of cells (increasing the amount of epitope) or decreasing staining volume (effectively reducing the amount of antibody without changing its concentration). For antibodies optimized to reach their 'saturation plateau' (common in flow cytometry), both of these approaches can be applied without changing the true signal. In contrast, for oligo-conjugated antibodies used in sequencing-based single-cell approaches, operating in the linear range, signal from highly abundant epitopes stained with low concentration of antibody will be affected. In such cases, the cells can be stained in multiple steps adjusting the staining volume while keeping the concentration the same – that is, staining in a smaller volume for antibodies with high background and subsequently staining antibodies at low concentration in a higher volume. In this regard, when multiplexing samples, pre-staining each sample with hashtags and pooling prior to staining with additional CITE-seq antibodies may provide multiple advantages: (1) all samples are stained at the same time with the exact same antibody mixture – making cross-sample comparison more accurate, (2) by having more cells in a smaller total volume, less total antibody is used in the presence of more epitopes conceivably reducing the background signal and (3) samples where cell number at staining is a limiting factor, such as small tissue biopsies, will be exposed to the same local concentrations of antibody as more abundant samples (such as PBMCs) removing potential differences between samples by antibodies being 'sponged' by differences in overall epitope abundance. However, this approach is only available when all samples are similarly affected by the staining procedure and can tolerate the additional washes needed (after both hashing and CITE-seq staining).

We compared ADT signal from PBMCs stained with the same antibody panel at the starting concentration with a sample stained at concentrations adjusted following the titration experiment. While some markers could benefit from further adjustments, the sample stained with the adjusted panel was more balanced in its distribution of sequencing reads among markers, having twice the median UMIs per positive cell. Despite intentionally reducing signal in category B antibodies, we found an overall 57% increase in the median positive signal. Concomitantly, the adjusted panel exhibited 43% lower background signal (median of 26.3% to 14.9% UMIs assigned to background) despite increasing the concentrations of many category C and E antibodies. Consequently, the adjusted concentrations greatly improved the overall performance of the panel. We took precautions to make the samples as comparable as possible by down-sampling the sequencing depth to the same level and comparing similar numbers of analogouscells (at the mRNA level). Nonetheless, as these samples were from different preparations and different donors, we cannot exclude that some of the observed differences can be attributed to these factors. For instance, we found that the monocytes in the adjusted sample exhibited higher nonspecific binding (as seen from the isotype controls) than in the DF1 sample, despite being treated with the same concentrations of Fc-blocking reagents (which should minimize such biding; *Andersen et al., 2016*).

Due to the 10- to 1000-fold higher numbers of individual proteins as compared to mRNA (*Marguerat et al., 2012*), ADT libraries have high library complexity (unique UMI content) and are rarely sequenced near saturation. Thus, either sequencing deeper or squandering less reads on a few antibodies increases signal from all (other) included antibodies. We found that by simply reducing the concentration of the five antibodies used at 10 µg/mL, we gained 17% more reads for the remaining antibodies. Consequently, assuming we are satisfied with the magnitude of signal we got from all other antibodies using the starting concentration, this directly translates to a 17% reduction in sequencing costs. Due to different antibodies being adjusted in different directions for different reasons (according to their assigned categories), it is difficult to convert the overall improved utilization of sequencing reads into exact savings calculation. However, assuming signal is improved or unchanged, the savings on sequencing for each marker can be estimated by how many UMIs are needed to acquire a given signal. In the case of CD86, we found that the signal was dramatically

improved by reducing concentration from 10 to 0.667 µg/mL while also using 79% fewer UMIs and consequently a much lower number of sequencing reads.

Empty droplets have been shown to be useful for determining the background signal of CITE-seq (*Mulè et al., 2020*). This suggests that the major source of background signal for ADT libraries can be attributed to free-floating antibodies (or oligos) in the solution rather than nonspecific antibody binding to cell surfaces. In the present study, the samples were multiplexed by hashing antibodies, pooled after oligo-conjugated antibody staining, and then run in the same 10X Chromium lane. This obscures the contribution of each sample to the total amount of free-floating antibodies in the final cell suspension, which is conceivably skewed towards the samples stained in high volume with the highest concentration of antibodies. Consequently, as free-floating antibodies are the major source of background, this would explain why we do not observe reduced background in the cells stained at the lowest concentrations (i.e., DF4). As such, for markers with no specific signal due to high background (such as CD183, CD197, and TCRgd), the titration responses may be underestimated due to specific signal being lost within the high background. This also means, that for markers with high background signal our proposed reductions in concentrations are conservative as we would expect to see decreased background in samples stained with reduced amount of antibodies (as seen in the comparison with the adjusted concentrations). In droplet-based single-cell analyses, background signal is not only diminishing the sensitivity and resolution of true signals, but is also a major contributor to sequencing cost of ADT libraries. Due to empty droplets vastly outnumbering cell-containing droplets, we found that ADT signal from empty droplets can easily account for 20–50% of the total sequencing reads and consequently 20–50% of the sequencing cost. The number of antibodies used in CITE-seq-related platforms is only expected to expand. Additionally, the number of cells included in each experiment is continuously being increased (as seen for methods such as SCITO-seq; *Hwang, 2020*). As such, reducing background signal from oligo-conjugated antibodies should be a priority. The source of the free-floating antibodies is not completely understood. Observations from this study suggest that antibodies used at high concentration targeting absent or sparse epitopes are highly enriched within the empty droplets, as compared to the cell-containing droplets. This indicates that residual unbound antibody from the staining step is a major contributor despite several washing steps. Practically, this suggests that additional washing after cell staining would be beneficial when the number and type of cells in the samples allow it. Optimal washing is achieved by repeated washing steps while assuring that maximal residual supernatant is removed after each centrifugation and followed by gentle but complete resuspension in a large buffer volume.

More and more advanced CITE-seq-related cytometry-by-sequencing platforms are rapidly being developed. However, while these platforms utilize different methods to assure single-cell resolution and use different approaches to label the cells, they all use high-throughput sequencing to count signal from a variety of oligo-conjugated probes (such as antibodies with both surface and intracellular targets, MHC-peptide multimers, and B-cell receptor antigens) (*Stoeckius et al., 2017*; *Peterson et al., 2017*; *Hwang, 2020*; *Setliff et al., 2019*; *O'Huallachain et al., 2020*; *Overall et al., 2020*; *Gaublomme et al., 2019*; *Katzenelenbogen et al., 2020*). Most of the observations and conclusions from this study will be applicable to tthese platforms, where improving oligo-conjugated probe signal is critical to their utility and economic feasibility.

## Materials and methods

### Clinical samples

Lung adenocarcinoma patient sample (female, 57 years old, former smoker: 15 pack-years, treated with chemotherapy) was collected at New York University Langone Health Medical Center in accordance with protocols approved by the New York University School of Medicine Institutional Review Board and Bellevue Facility Research Review Committee (IRB#: i15-01162 and S16-00122).

### Cell isolation, cryopreservation, and thawing

PBMCs were isolated from a leukopak and whole blood from healthy donors (New York Blood Center) for the pre-titration and adjusted samples, respectively. PBMCs were purified by diluting in PBS and subsequent gradient centrifugation using Ficoll-Paque PLUS (GE Healthcare) and 50 mL conical tubes (Falcon). PBMCs in the interphase were collected and washed twice with PBS containing 2%

FBS. Lung tumor sample were cut into small pieces with a razor blade and enzymatically digested (100 U/mL Collagenase IV, Sigma-Aldrich, C5138-1G; 50 µg/mL DNase 1, Worthington, LS002138) for 35 min being rotated at 37°C in HEPES buffered RPMI 1640 containing 0.5% FBS. After digestion, the sample was forced through a 100 µm cell strainer to make a single-cell suspension. Single-cell suspensions from both PBMCs and lung tumor were cryopreserved in freezing medium (40% RPMI 1640, 50% FBS, and 10% DMSO) and stored in liquid nitrogen. On the day of the experiments, cryo-preserved samples were thawed for 1–2 min in a 37°C water bath, washed twice in warm PBS containing 2% FBS, and resuspended in complete media (RPMI 1640 supplemented with 10% FBS and 2 mM L-Glut).

## Oligo-conjugated antibody staining

We modified the published protocol for ECCITE-seq (*Mimitou et al., 2019*) to stain cells in round-bottom 96-well plates (as is common practice for flow cytometry staining in many laboratories). This allowed us to reduce staining volumes and centrifugation time analogous to staining for flow cytometry. After thawing, the intended number of cells was resuspended in 12.5 µL or 25 µL of CITE-seq staining buffer (2% BSA, 0.01% Tween in PBS) for samples stained in a total of 25 µL or 50 µL, respectively. To prevent antibody binding to Fc receptors (*Andersen et al., 2016*), Fc receptor block from two vendors (TruStain FcX, BioLegend, and FcR blocking reagent, Miltenyi) was added to the suspension and incubated for 10 min on ice. During incubation, the antibody solution of 52 Total-SeqC antibodies (BioLegend; *Supplementary file 1*) was washed on a pre-wet Amicon Ultra-0.5 Centrifugal Filter to remove sodium azide. The volume of the resulting antibody pool was adjusted to 2× of final concentrations and 12.5 µL or 25 µL was added to the cells to achieve a total staining volume of 25 µL or 50 µL, respectively. 10 µg/mL of a unique hashing antibody was added to each sample and incubated for 30 min on ice. After staining, cells were washed four times in 1 × 150 µL and 3 × 200 µL CITE-seq staining buffer.

## Super-loading of 10X Chromium

Individually hashed samples were counted using a hemocytometer and pooled in equal ratio at high concentration. Pooled sample was strained through a 70 µm cell strainer and counted again using a hemocytometer. To achieve approximately 20,000 cells after doublet removal, cell concentration was adjusted to 1314 cells/µL to achieve the target of 41,645 cells in 31.7 µL for super-loading of the 10X Chromium Chip A. Gene expression (using 5′ v1 chemistry; 10X Genomics) and ADT and hashtag-oligo (HTO) libraries were constructed using reagents, primers, and protocol from the published ECCITE-seq protocol (*Mimitou et al., 2019*). All libraries from the titration run were sequenced together with other samples on an Illumina NovaSeq6000 S1 flow cell. The post-titration (adjusted) sample (using 5′ v1.1 chemistry; 10X Genomics) was multiplexed and sequenced together with other samples not included in this study on Illumina NovaSeq6000 SP and S1 flow cells.

## Alignment and counting of single-cell sequencing libraries

The multiplexed gene expression library was aligned using kallisto (v0.46)-bustools (v0.39.0) (*Melsted et al., 2021*). Given the polyA selection inherent in the 10X Genomics protocol, reads were aligned against a reference transcriptome based on the GTF file included in the Cell Ranger software (refdata-cellranger-GRCh38-3.0.0/genes/genes.gtf; 10X Genomics) that does not include as many non-polyA transcripts as the human transcriptome included by kallisto-bustools by default. From the 77,507,446 reads assigned to the gene expression library, 66.9% aligned to the transcriptome. ADT and HTO libraries were counted using the *kallisto indexing and tag extraction* (KITE) workflow (https://github.com/pachterlab/kite), resulting in 82,527,351 and 65,875,774 counted reads, respectively. Number of UMIs and genes detected per cell across cell lineages can be found in *Figure 1—figure supplement 1A, B*.

## Single-cell demultiplexing, preprocessing, and down-sampling

To allow detection of UMI counts within non-cell-containing droplets, unfiltered count matrices from each modality were loaded into a 'Seurat' (v3.1.4) object (*Stuart et al., 2019*). Samples were demultiplexed by their unique HTO using the Seurat function 'MULTIseqDemux' yielding 19,560 demultiplexed cells. This allowed the removal of 3724 (19%) cross-sample doublets. Due to the shallow

sequencing of the mRNA library (~4000 reads/cell), expression of at least 60 genes and a percent mitochondrial reads below 15% were used to remove barcodes from non-viable cells or debris (2499 or 15% of cells removed). Intra-sample doublets were removed using the 'scDblFinder' (v1.1.8) R package (392 cells removed). UMI counts from ADTs were normalized using default configuration of the DSB (v0.1.0) R package with ADT signal from HTO-negative droplets used as empty drop matrix and using included isotype controls. Gene expression was preprocessed using the default Seurat v3 pipeline, and fine-grained clusters were identified using the 'FindClusters' function with a resolution of 1.2. Clusters were annotated by lineages and cell types using their distinct expression of markers within the mRNA or ADT modality and aided by cell-by-cell annotation from the SingleR R package (v1.4.0) using the 'Monaco reference' from the celldex R package (v1.0.0) made from bulk RNA-seq samples of sorted immune cell populations from GSE107011 (*Monaco et al., 2019*). Top five differentially expressed marker genes for each cluster can be found in *Figure 1—figure supplement 1D*. To allow direct comparison of UMI counts across conditions, each condition was down-sampled by tissue of origin to include the same number of cells within each fine-grained cell-type cluster (resulting in 1777 cells from each PBMC sample and 1681 cells from each lung tumor sample).

### Integration and sub-sampling for pre- and post-titration comparison

The post-titration (adjusted) sample was pre-processed as described above. Together with the DF1 sample, the adjusted sample was normalized and integrated based on their mRNA expression using the SCTransform and IntegrateData functions from the Seurat package as described in the Seurat integration vignette (*Stuart et al., 2019*; *Hafemeister and Satija, 2019*). After mRNA-based clustering using FindClusters at resolution 1.2, similar number of comparable cells was selected by taking the nearest neighbors in PCA-space for each cell in the sample with the fewest cells within the given cluster. This sampling assured that similar number of comparable cells (at the mRNA level) were selected for comparison, thus minimizing the effect of the sample differences. To allow direct comparison of UMI counts and eliminate differences in sequencing depth as a factor, we down-sampled the FASTQ files from the ADT modality of the adjusted sample to achieve similar totals of UMIs within the DF1 (522,469) and adjusted (521,331) samples.

### Comparing ADT signal from cell-containing and empty droplets

For comparison of UMI counts within cell-containing and non-cell-containing (empty) droplets for the present dataset and the 10X Genomics datasets, we divided the unfiltered count matrices by the inflection point in their ranked per cell UMI sum from the mRNA library. Barcodes above the inflection point were then used to extract UMI counts within cell-containing droplets from each antibody oligo modality. All UMIs that were not included in cell-containing droplets were considered from empty droplets.

### Data and code availability

All codes and commands used to process the data and generate all plots and figures are available at GitHub: https://github.com/Terkild/CITE-seq_optimization (*Buus, 2021*; copy archived at swh:1:rev:1c7fcabb18a1971dc4d6e29bc3ed4f6f36b2361f).

UMI count matrices from the optimization experiment have been deposited at FigShare with DOI: https://doi.org/10.6084/m9.figshare.c.5018987. The feature barcode 3′ and 5′ VDJ 10X datasets are available from the 10X Genomics website.

## Acknowledgements

Work in Dr. Koralov's laboratory was supported by the NIH R01 grant (HL-125816), LEO Foundation Grant (LF-OC-20–000351), NYU Cancer Center Pilot Grant (P30CA016087), the Judith and Stewart Colton Center for Autoimmunity Pilot grant, and a grant from the Drs. Martin and Dorothy Spatz Foundation. TBB and NØ are supported by the Danish Cancer Society (Kræftens Bekæmpelse), the Danish Council for Independent Research (Danmarks Frie Forskningsfond), and the LEO Foundation. We thank the NYU Genome Technology Center for technical assistance and support and acknowledge the NYU Center for Biospecimen Research and Development and NYU Perlmutter Cancer Center for their support in acquiring patient biospecimens.

# Additional information

## Competing interests

Peter Smibert: is co-inventor of a patent related to the single cell technology utilized in this study (US provisional patent application 62/515-180). The other authors declare that no competing interests exist.

## Funding

| Funder | Grant reference number | Author |
|---|---|---|
| National Institutes of Health | HL-125816 | Sergei B Koralov |
| LEO Pharma Research Foundation | LF-OC-20-000351 | Niels Odum<br>Sergei B Koralov |
| NYU School of Medicine | P30CA016087 | Sergei B Koralov |
| Judith and Stewart Colton Center for Autoimmunity Pilot Grant | | Sergei B Koralov |
| Drs. Martin and Dorothy Spatz Foundation | | Sergei B Koralov |
| Kræftens Bekæmpelse | | Terkild Brink Buus<br>Niels Odum |

The funders had no role in study design, data collection and interpretation, or the decision to submit the work for publication.

## Author contributions

Terkild B Buus, Conceptualization, Data curation, Software, Formal analysis, Investigation, Visualization, Methodology, Writing - original draft, Project administration, Writing - review and editing; Alberto Herrera, Ellie Ivanova, Conceptualization, Investigation, Writing - original draft, Writing - review and editing; Eleni Mimitou, Peter Smibert, Conceptualization, Resources, Methodology, Writing - review and editing; Anthony Cheng, Software, Visualization, Writing - review and editing; Ramin S Herati, Resources, Data curation, Funding acquisition, Writing - review and editing; Thales Papagiannakopoulos, Conceptualization, Writing - review and editing; Niels Odum, Conceptualization, Supervision, Funding acquisition, Writing - review and editing; Sergei B Koralov, Conceptualization, Funding acquisition, Methodology, Writing - original draft, Project administration, Writing - review and editing

## Author ORCIDs

Terkild B Buus https://orcid.org/0000-0001-7180-6384
Alberto Herrera http://orcid.org/0000-0003-4189-9051
Ellie Ivanova https://orcid.org/0000-0002-1850-9505
Eleni Mimitou https://orcid.org/0000-0001-9737-6394
Anthony Cheng https://orcid.org/0000-0002-0778-8238
Ramin S Herati https://orcid.org/0000-0003-2613-4050
Thales Papagiannakopoulos http://orcid.org/0000-0002-2251-1624
Peter Smibert https://orcid.org/0000-0003-0772-1647
Niels Odum http://orcid.org/0000-0003-3135-5624
Sergei B Koralov https://orcid.org/0000-0002-4843-3791

## Ethics

Human subjects: Lung adenocarcinoma patient sample was collected at New York University Langone Health Medical Center in accordance with protocols approved by the New York University School of Medicine Institutional Review Board and Bellevue Facility Research Review Committee (IRB#: i15-01162 and S16-00122).

Decision letter and Author response
Decision letter https://doi.org/10.7554/eLife.61973.sa1
Author response https://doi.org/10.7554/eLife.61973.sa2

## Additional files

### Supplementary files

• Supplementary file 1. Antibody panel and concentrations. Table of the 52 antibodies included in the panel. Also contains individual clones and concentrations used for the different conditions included in the study.

• Supplementary file 2. Antibody cost calculations. Antibody costs of the 52 antibody panel using vendor recommendations for staining volume and concentrations, pre-titration (dilution factor 1) concentrations, and adjusted concentrations.

• Transparent reporting form

### Data availability

All code and commands used to process the data and to generate all plots and figures are available at GitHub: https://github.com/Terkild/CITE-seq_optimization and a copy is archived at https://archive.softwareheritage.org/swh:1:rev:1c7fcabb18a1971dc4d6e29bc3ed4f6f36b2361f/. UMI count matrices from the optimization experiment have been deposited at FigShare with https://doi.org/10.6084/m9.figshare.c.5018987. The feature barcode 3' and 5' VDJ 10X datasets are available from the 10X Genomics website.

The following dataset was generated:

| Author(s) | Year | Dataset title | Dataset URL | Database and Identifier |
|---|---|---|---|---|
| Buus TB, Herrera A, Ivanova E, Mimitou E, Cheng A, Herati R, Papagiannakopoulos T, Smibert P, Ødum N, Koralov SB | 2020 | Improving oligo-conjugated antibody signal in multimodal single-cell analysis | https://doi.org/10.6084/m9.figshare.c.5018987 | figshare, 10.6084/m9.figshare.c.5018987.v1 |

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
