## [Decision Letter]

**Acceptance summary:**

The work teaches us how to optimize oligo-conjugated antibodies for droplet-based scRNA-seq studies (CITE-seq). This study provides a careful assessment of oligo-conjugated antibody signal in CITE-seq, testing several relevant variables, and clearly demonstrating that antibody titration is a crucial step to optimize a CITE-seq panel.

**Decision letter after peer review:**

Thank you for submitting your article "Improving oligo-conjugated antibody signal in multimodal single-cell analysis" for consideration by *eLife*. Your article has been reviewed by 2 peer reviewers, and the evaluation has been overseen by Michael Eisen as the Senior and Reviewing Editor. The following individuals involved in review of your submission have agreed to reveal their identity: José Ordovas-Montanes (Reviewer #1); Johan Duchene (Reviewer #2).

The reviewers have discussed the reviews with one another and the Reviewing Editor has drafted this decision to help you prepare a revised submission.

Summary:

In the study by Buus et al., the authors set out to address an important need to understand how oligo-conjugated antibodies should be optimally utilized in droplet-based scRNA-seq studies. These techniques, often referred to as CITE-seq, complement techniques such as flow cytometry and mass cytometry yet also further extend them by the ability to jointly measure intra-cellular RNA-based cell states together with antibody-based measurements. As is the case with flow cytometry, manufacturers provide staining recommendations, yet encourage users to titrate antibodies on their specific samples in order to derive a final staining panel. Based on the ability to stain with hundreds of antibodies jointly, few studies to date have assessed how the antibodies present in these pre-made staining panels respond to a standard titration curve. In order to address this point, this study tests two dilution factors, staining volume, cell count, and tissue of origin to understand the relationships between signal and background for a commercially available antibody panel. They arrive at the general recommendation that these panels could be improved, grouping various antibodies into distinct categories.

This study is of general interest to the scRNA-seq and CITE-seq communities as it draws attention to this important aspect of CITE-seq panel design. However, given the title is "improving oligo-conjugated antibody in multi-modal single cell analysis", the manuscript would be substantially improved by not only providing suggestions but also testing at least one, if not more, of their suggestions from Supplementary Table 2, and preferably performing experiments using more technical replicates or biological replicates. As it stands now, the study is largely based on one PBMC and one lung sample, that were stained once with each manipulation as far as can be gathered from the Methods.

Recombinant antibodies are the most common and powerful reagents in life science research to identify and study proteins. Yet, every single antibody should always be validated and carefully tested for its relevant application, to ensure constructive and reproductive scientific endeavor. I was thus extremely pleased to review the manuscript of Terkild Buus et al., as it provides a careful assessment of oligo-conjugated antibody signal in CITE-seq. The authors tested four variables (antibody concentration, staining volume, cell numbers and tissue origin) and clearly showed that antibody titration is a crucial step to optimize CITE-seq panel. The authors found that, as a general rule, concentration in the 0.625 and 2.5 µg/mL range provides the best results while recommended concentrations by vendors, 5 to 10 µg/mL range, increase background signal.

Essential revisions:

1. The starting concentration used for each antibody was based on historical experience and assumptions about the abundance of the epitopes. This approach may not be ideal, and the optimal concentration may have been missed. Do the authors think that a proper titration would be an advantage? Maybe this could be discussed in the text.As a means of testing, we suggest a full titration curve of selected antibodies, perhaps one from each of the categories, but if cost is a concern at least two or three antibodies, to identify how titration impacts antibodies, and especially those in categories labeled as in need of improvement. Relatedly, if the idea is that if antibodies (such as gD-TCR) do not have a cognate receptor leading to general background spread, does spiking in a cell that is a known positive in increasing ratios remedy this issue by acting as a target for the antibodies? Does adding extra washes help to remedy these issues of background?

2. Another way of improving these panels is through reducing the costs spent on both staining but perhaps more importantly the sequencing-based readouts. Several times in the manuscript (at line 77 for example or line 277) it is alluded to that the background signal of antibodies can make up a substantial cost of sequencing these libraries. However, no formal data on cost is presented, which would be important to formalize the author's points. It would be important to provide cost calculations and recommendations on sequencing depth of ADT libraries based on variation of staining concentration. Relatedly, in the methods, sequencing platform and read depth for ADT libraries was not discussed, nor is the RNA-seq quality control metrics provided other than a mention of ~5,000 reads/cell targeted. This is important to report in all transcriptomic studies, and especially a methods development study.

3. One of the powerful elements of joint multi-modal profiling, as mentioned in the title, is to be able to measure protein and RNA from a single cell. This study does not formally look at correlation of protein and RNA levels, and whether a decrease in concentration of antibody either improves or diminishes this correlation. This would be important to test with in this study to ensure that decreasing antibody levels does not then adversely affect the power of correlating protein with RNA, and whether it may even improve it.

Relatedly, the authors showed by testing four variables (see above) that they could define the optimal conditions to reduce background signal and increase sensitivity of antibodies and thus this way improves CITE-seq outcome. Nevertheless, the authors rely on the fact that all antibodies used in their panel are specific for their targeted antigens. It is not necessary to experimentally test the specificity of every single antibody used in the study as this would be a colossal amount of work. But I feel that this aspect should be discussed in the manuscript, especially when an "uncommon" antibody is intended to be used in the CITE-seq panel; the specificity of this antibody should be indeed tested prior to its use.

4. How was the lack of antibody binding determined for Category E? CD56 is frequently detected on NK cells in peripheral blood, CD117 should be detected on mast cells in lung, and CD127 should be found on T cells, particularly CD8^+^ T cells. From inspecting Figure 1E, it appears as if all three of these markers are detected on small but consistent cell subsets. As the clusters are only numbered and no supplementary table is provided to help to reader in their interpretation, it is difficult to determine if these represent rare but specific binding, or have not bound with any specificity.

5. References: At 14 references, the paper overall could benefit from a more comprehensive citation of related literature including flow cytometry and/or CyTOF best practices for antibody staining and dealing with background, and joint RNA and protein measurement from single cells.

---

## [Author Response]

Summary:In the study by Buus et al., the authors set out to address an important need to understand how oligo-conjugated antibodies should be optimally utilized in droplet-based scRNA-seq studies. These techniques, often referred to as CITE-seq, complement techniques such as flow cytometry and mass cytometry yet also further extend them by the ability to jointly measure intra-cellular RNA-based cell states together with antibody-based measurements. As is the case with flow cytometry, manufacturers provide staining recommendations, yet encourage users to titrate antibodies on their specific samples in order to derive a final staining panel. Based on the ability to stain with hundreds of antibodies jointly, few studies to date have assessed how the antibodies present in these pre-made staining panels respond to a standard titration curve. In order to address this point, this study tests two dilution factors, staining volume, cell count, and tissue of origin to understand the relationships between signal and background for a commercially available antibody panel. They arrive at the general recommendation that these panels could be improved, grouping various antibodies into distinct categories.

We appreciate the reviewers insight into the methodology and enthusiasm for the study. We do want to clarify that the study does not use a “pre-made staining panel” from commercial vendor, but rather a cocktail of individual antibodies available from a commercial vendor (with emphasis on epitopes relevant to immunology and cancer research). We have also clarified this in the text of the manuscript.

This study is of general interest to the scRNA-seq and CITE-seq communities as it draws attention to this important aspect of CITE-seq panel design. However, given the title is "improving oligo-conjugated antibody in multi-modal single cell analysis", the manuscript would be substantially improved by not only providing suggestions but also testing at least one, if not more, of their suggestions from Supplementary Table 2, and preferably performing experiments using more technical replicates or biological replicates. As it stands now, the study is largely based on one PBMC and one lung sample, that were stained once with each manipulation as far as can be gathered from the Methods.

We hope that the added analysis, our point by point response to the issues raised by the reviewer, and inclusion of new CITE-seq data from the panel with adjusted concentrations to alleviates the main concerns of the reviewers. We appreciate the suggestions and believe that we now present a much-improved manuscript for your review.

Recombinant antibodies are the most common and powerful reagents in life science research to identify and study proteins. Yet, every single antibody should always be validated and carefully tested for its relevant application, to ensure constructive and reproductive scientific endeavor. I was thus extremely pleased to review the manuscript of Terkild Buus et al., as it provides a careful assessment of oligo-conjugated antibody signal in CITE-seq. The authors tested four variables (antibody concentration, staining volume, cell numbers and tissue origin) and clearly showed that antibody titration is a crucial step to optimize CITE-seq panel. The authors found that, as a general rule, concentration in the 0.625 and 2.5 µg/mL range provides the best results while recommended concentrations by vendors, 5 to 10 µg/mL range, increase background signal.Essential revisions:1. The starting concentration used for each antibody was based on historical experience and assumptions about the abundance of the epitopes. This approach may not be ideal, and the optimal concentration may have been missed. Do the authors think that a proper titration would be an advantage? Maybe this could be discussed in the text.As a means of testing, we suggest a full titration curve of selected antibodies, perhaps one from each of the categories, but if cost is a concern at least two or three antibodies, to identify how titration impacts antibodies, and especially those in categories labeled as in need of improvement. Relatedly, if the idea is that if antibodies (such as gD-TCR) do not have a cognate receptor leading to general background spread, does spiking in a cell that is a known positive in increasing ratios remedy this issue by acting as a target for the antibodies? Does adding extra washes help to remedy these issues of background?

These are excellent points. We agree that using starting concentrations based on historical experience etc. may not be ideal for a completely objective assessment of how oligo-conjugated antibodies respond to the four-variables test. However, we firmly believe that using informed starting concentrations greatly increases the potential improvement of a panel while keeping costs to a minimum (which has to be a consideration for these expensive methods). With that said, we agree that this approach may not reach *the* optimal concentration (a definition that is a bit complex in this setting). Full titration curves have previously been published showing that oligo-conjugated antibodies respond to titration, and in that regard behave similar to fluorophore-conjugated antibodies assayed by flow cytometry (see Stoeckius et al. 2018. Genome Biology; Figure 3A-D). Our study does not aim to identify the optimal concentration of individual antibodies in isolation but strives to provide the optimal signal-to-noise ratio for each antibody in a cocktail while taking sequencing requirements into account – this is why we don’t focus on full titration curves and saturation kinetics for each antibody/epitope. If we use all antibodies at their highest signal-to-noise ratios, this would drastically increase sequencing requirements of the library as highly expressed markers would use the vast majority of the total sequencing reads. As such, we aimed to get “sufficient” signal-to-noise while keeping the sequencing allocated to each marker balanced. We have elaborated on this in the discussion of the revised manuscript.

Furthermore, as our results show, background signal can be largely attributed to free-floating antibodies in the solution, using high concentrations for all markers in one or more condition would increase the background in all conditions if these were multiplexed into the same droplet segregation. This phenomenon would likely obscure the positive signals and possibly titration response at lower concentrations (similar to what we see for category A antibodies). To avoid this, if full titration curves should be meaningful, each condition should be run in its own droplet segregation making such titration efforts prohibitively costly. We have elaborated on this in the discussion of the revised manuscript.

We agree that it would greatly improve the study to include results from our panel with adjusted concentrations. In the revised manuscript, we have made efforts to address this by making a comparison between the sample stained with the pre-titration (DF1) concentrations and a sample stained with concentrations that have adjusted based on their assigned categories (from Table 1). We believe that this new data convincingly demonstrates improvements both of the individual antibody signals and at the level of the increased sequencing balance (see new Figure 5). While the adjusted concentrations could still benefit from further improvements, we show that at similar sequencing depths, the adjusted concentrations provide a more balanced sequencing output and exhibit a 57 % increase in the median positive signal and a 43 % reduction in the median background signal for the 52 antibodies in our panel. The benefit of the adjusted concentration was particularly remarkable for CD86 which went from having 76.5 % to 12.6 % of UMIs assigned to background signal and thus yielded comparable positive signal while using 4.8 fold less UMIs (new Figure 5G).

Spiking in cells that express the cognate antigen is an interesting idea. However, as the spiked in cells would be included in all the downstream processes including sequencing of mRNA and other modalities, it would be quite costly to spike-in cells that are not of biological interest – only to decrease background of one or a few antibodies.

While the results presented in the manuscript do not address this directly, our data strongly suggest that adding extra washing would help reduce free-floating antibodies in the solution captured in the gel-bead emulsions responsible for some of the observed background signal (as can be assayed by the non-cell-containing droplets). For such a test to make sense, the staining conditions should be identical for two samples that are differentially washed (including the exact same cell composition) and would require fully separate droplet segregations (i.e. utilization of separate 10x lanes) which would make it a very costly experiment solely to test the washing effect. However, we have done preliminary tests using short (150bp) cDNA amplicon spiked into different tubes or plates containing ~750x10^3^ PBMCs to determine washing efficiency by qPCR. Here we assayed how increasing the washing volume from 200µl (96-well) to 1.5mL or 50mL for two washes reduced the detection of the spiked-in amplicon in the supernatant as compared to an unwashed sample. While short cDNA amplicons may not behave identical to oligo-conjugated antibodies, they simulate background signal stemming from free-floating antibodies and thus can be used to evaluate different washing conditions for a given set-up. As expected, using higher washing volumes does indeed greatly reduce the amount of amplicon (simulating free-floating “background” antibodies) detected in the resulting suspension.

2. Another way of improving these panels is through reducing the costs spent on both staining but perhaps more importantly the sequencing-based readouts. Several times in the manuscript (at line 77 for example or line 277) it is alluded to that the background signal of antibodies can make up a substantial cost of sequencing these libraries. However, no formal data on cost is presented, which would be important to formalize the author's points. It would be important to provide cost calculations and recommendations on sequencing depth of ADT libraries based on variation of staining concentration. Relatedly, in the methods, sequencing platform and read depth for ADT libraries was not discussed, nor is the RNA-seq quality control metrics provided other than a mention of ~5,000 reads/cell targeted. This is important to report in all transcriptomic studies, and especially a methods development study.

Thank you for pointing out the very sparse description of choice of sequencing method and RNA-seq quality controls. We have included additional metrics in the Materials and methods and included a new Figure 1—figure supplement 1 showing number of detected genes as well as UMI counts within the mRNA and ADT modalities in the revised manuscript. We agree that reducing sequencing cost (without reducing biological information) is a major reason for optimizing staining with oligo-conjugated antibodies. We have now added a section in which we elaborate on the potential cost saving, and other benefits of titration of antibody panels and provide some examples from our datasets. Actual savings of optimization of these panels will be very dependent on a given setup, starting concentrations and the depth of sequencing that the particular research questions (and budget) warrant.

Due to the 10-1000 fold higher numbers of proteins as compared to coding mRNA [16], ADT libraries have high library complexity (unique UMI content) and are rarely sequenced near saturation. Thus, either sequencing deeper or squandering fewer reads on a handful of antibodies, will result in an increased signal from other antibodies in the panel. We found that by simply reducing the concentration of the five antibodies used at 10 µg/mL, we gained 17 % more reads for the remaining antibodies. Consequently, assuming we are satisfied with the magnitude of signal we got from all other antibodies using the starting concentration, this directly translates to a 17 % reduction in sequencing costs.

In terms of sequencing depth, we are not comfortable giving very broad recommendations. This is due to the fact that sequencing requirements will be very different depending on the composition of the antibody panel as well as the cell type distribution (epitope abundance) (as has been previously noted in Mair et al. 2020 Cell Rep.). If the antibody panel contains only antibodies targeting epitopes that are largely present on a small subset of cells (such as CD56 or CD8 for PBMCs) it would require fewer reads per marker per total cell count than markers that are broadly expressed (such as HLA-ABC or CD45 for PBMCs). However, in a different sample composition (for instance a tissue with few leukocytes) these same antibodies would require fewer reads per cell whereas other epitopes may be more abundant.

We want to also stress, that aside from cost savings, an optimized balanced panel with low background will yield improved resolution compared to a non-optimized panel. Fortunately, CITE-seq and related methods are very flexible in this regard as you can start by shallow sequencing and then “top-up” the sequencing depth to an optimal level based on the actual data in subsequent sequencing runs (for instance together with the next batch of samples).

3. One of the powerful elements of joint multi-modal profiling, as mentioned in the title, is to be able to measure protein and RNA from a single cell. This study does not formally look at correlation of protein and RNA levels, and whether a decrease in concentration of antibody either improves or diminishes this correlation. This would be important to test with in this study to ensure that decreasing antibody levels does not then adversely affect the power of correlating protein with RNA, and whether it may even improve it.

We appreciate the reviewer’s suggestion – this is a great idea. Unfortunately, such correlations are notoriously hard to do for scRNA-seq data due to the sparsity of the RNA measurements (which contains high frequency of 0 UMI counts). This is, in part, due to low reverse transcriptase efficiency, and also due to the fact that most proteins have 10-1000 fold more copies than the mRNA transcripts that encode them (Marguerat et al. 2012 Cell). This is exacerbated in our study by the fact that we only shallowly sequenced RNA modality (~4000 reads/cell). Consequently, we see a very high number of cells that despite clustering together within distinct lineage clusters (based on their full transcriptome) and expressing the expected lineage marker surface proteins, do not have readily detectable transcript for the same marker(s). For instance, for all cells that are positive for CD8 at the RNA level, there are at least as many that are negative for CD8 RNA while being positive for CD8 ADT. Importantly, these additional CD8^+^ cells are still located within clusters consistent with a CD8^+^ phenotype (see Author response image 2).

**Author response image 2. respfig2:** 

As such, due to the sparsity of RNA counts, if ADT signal is diluted too much leading to truly positive cells being called as negative, it may actually increase individual cell correlation between RNA and ADT but mean higher levels of “false negative” cells. Direct correlation between RNA and antibody measurements within each individual cells is further complicated by the presence of non-specific/background signal in protein data that is rarely found in RNA data. This can also be seen in the Author response image 2 by the fact that positive cells are defined at a cut-off “7” at the ADT level, and not “0” as is the case for RNA. Thus, while having only a few UMI counts for a given transcript is sufficient to call expression, having a few UMIs from an ADT can easily be attributed to background (particularly in an unoptimized panel).Due to these technical limitations, we find it more suitable to correlate “positivity” called by either ADT (gated positive as shown in Figure 1—figure supplement 2) or mRNA expression (i.e. > 0 UMI counts). While this comparison is less quantitative (does not distinguish “high” from “low” expression) it enables us to show whether reducing antibody concentrations affects ADT signal ability to distinguish positive from negative cells (as compared to GEX), which is at the core of the reviewer’s suggestion. Author response image 3, demonstrates that four-fold titration reduces the fraction of positive cells by some markers (reduction in the blue+red bars by dilution) whereas other markers are largely unaffected both of which is consistent with the analysis in the manuscript:

**Author response image 3. respfig3:** 

In terms of assuring specificity, we have also modified the “titration plots” to show more detailed cell type distribution at each rank (by the “barcode plot” to the right of the “rank plot”) as well as the distribution of UMIs among cell types (by the bar plot above the “barcode plot”) at each condition. Finally, to make these “titration plots” more accessible, we have now included a guide to the different components of the “titration plots” in Figure 2 of the revised manuscript.

Relatedly, the authors showed by testing four variables (see above) that they could define the optimal conditions to reduce background signal and increase sensitivity of antibodies and thus this way improves CITE-seq outcome. Nevertheless, the authors rely on the fact that all antibodies used in their panel are specific for their targeted antigens. It is not necessary to experimentally test the specificity of every single antibody used in the study as this would be a colossal amount of work. But I feel that this aspect should be discussed in the manuscript, especially when an "uncommon" antibody is intended to be used in the CITE-seq panel; the specificity of this antibody should be indeed tested prior to its use.

Thank you for this suggestion. This is indeed an aspect of antibody optimization that we have not touched upon. By using commercially available oligo-conjugated antibody clones that are broadly used, the extensive testing of many of these clones by multiple labs within immunology community (for flow/mass cytometry applications) and based on our personal experience with majority of the clones for flow cytometry applications, we expected that the antibodies in our panel should be specific for their antigen. This is supported by the labelling matching what we would expect to find in PBMCs and lung leukocytes, as well as the correlation between expression of the gene encoding the targeted epitope and antibody binding (see our response to reviewer 1, point 3). We have added a paragraph to the revised manuscript discussing that, particularly when using antibodies for the first time or using clones that are unfamiliar, it is important to assure specificity.

4. How was the lack of antibody binding determined for Category E? CD56 is frequently detected on NK cells in peripheral blood, CD117 should be detected on mast cells in lung, and CD127 should be found on T cells, particularly CD8^+^ T cells. From inspecting Figure 1E, it appears as if all three of these markers are detected on small but consistent cell subsets. As the clusters are only numbered and no supplementary table is provided to help to reader in their interpretation, it is difficult to determine if these represent rare but specific binding, or have not bound with any specificity.

Thank you pointing this out. In light of this comment, it is obvious that we need to annotate the cell types of the clusters. We have annotated all the fine-grained clusters by cell types and re-worked all relevant panels in Figures 1, 2 and 3 (and all their related figure supplements) to show more detailed and consistent cell type annotation. We have also added Figure 1—figure supplement 1C and D to show marker genes for each of the annotated cell types, which together with the re-worked Figure 1E, give the reader a clear description of the cluster identity. We do indeed see some signal for Category E antibodies such as CD56, CD117 and CD127 within the expected clusters. This indicates that the antibodies do work to some extent. However, we also find that the signal for these markers is modest, at best, and not present in some populations where we would have expected them (CD127 should be more pronounced in T cells and we are finding an unexpectedly high frequency of CD56-negative NK cells).

5. References: At 14 references, the paper overall could benefit from a more comprehensive citation of related literature including flow cytometry and/or CyTOF best practices for antibody staining and dealing with background, and joint RNA and protein measurement from single cells.

We agree that the reference list of the original manuscript was sparse and may have missed important relevant studies. We have done our best to include additional studies relevant for the optimization and titration of mass cytometry panels and flow cytometry staining and added references to a few newly published joint RNA and protein measurement studies. We have strived to reference all studies directly relevant to the present work and do not want to overlook any appropriate publications that should be referenced and so welcome any suggestions of the reviewers.